# Preimplantation Developmental Competence of Bovine and Porcine Oocytes Activated by Zinc Chelation

**DOI:** 10.3390/ani12243560

**Published:** 2022-12-16

**Authors:** Juan P. Cabeza, Juan Cámera, Olinda Briski, Minerva Yauri Felipe, Daniel F. Salamone, Andrés Gambini

**Affiliations:** 1Facultad de Agronomía, Universidad de Buenos Aires, Ciudad Autónoma de Buenos Aires, Buenos Aires C1417DSE, Argentina; 2CONICET-Universidad de Buenos Aires, Instituto de Investigaciones en Producción Animal (INPA), Ciudad Autónoma de Buenos Aires, Buenos Aires C1417DSE, Argentina; 3School of Agriculture and Food Sciences, The University of Queensland, Gatton, QLD 4343, Australia

**Keywords:** assisted oocyte activation, embryo development, oocyte to embryo transition, zinc chelator, calcium oscillation

## Abstract

**Simple Summary:**

Although naturally triggered by sperm, oocyte activation can be achieved artificially to improve the development of fertilized eggs, produce nuclear transfer embryos, and induce parthenogenetic development. Calcium and zinc are known to play an essential role in this process. Exploring the impact of oocyte artificial activation protocols on development across different mammalian models is critical before application to humans. Here, we report the production of bovine and porcine blastocysts by using a novel zinc chelator. Developmental rates and the expression of key transcription factors were compromised in bovine when using the zinc chelator compared to calcium-induced embryos. On the contrary, a new concentration and incubation time of the zinc chelator allowed higher developmental rates without affecting embryo quality in porcine. Our results contribute to understanding the role of zinc during oocyte activation and preimplantation embryo development across different mammalian species.

**Abstract:**

After sperm-oocyte fusion, intracytoplasmic rises of calcium (Ca) induce the release of zinc (Zn) out of the oocyte (Zn sparks). Both phenomena are known to play an essential role in the oocyte activation process. Our work aimed to explore different protocols for activating bovine and porcine oocytes using the novel zinc chelator 1,10-phenanthroline (PHEN) and to compare developmental rates and quality to bovine IVF and parthenogenetic ionomycin-induced embryos in both species. Different incubation conditions for the zinc chelator were tested, including its combination with ionomycin. Embryo quality was assessed by immunofluorescence of SOX2, SOX17, OCT4, and CDX2 and total cell number at the blastocyst stage. Even though blastocyst development was achieved using a zinc chelator in bovine, bypassing calcium oscillations, developmental rates, and blastocyst quality were compromised compared to embryos generated with sperm-induced or ionomycin calcium rise. On the contrary, zinc chelation is sufficient to trigger oocyte activation in porcine. Additionally, we determined the optimal exposure to PHEN for this species. Zinc chelation and artificial induction of calcium rise combined did not improve developmental competence. Our results contribute to understanding the role of zinc during oocyte activation and preimplantation embryo development across different mammalian species.

## 1. Introduction

Zinc homeostasis orchestrates diverse cellular processes in most organisms since it is essential for the activity of many enzymes, transcription factors, and other signaling molecules. In consequence, alterations in the physiological concentration of this ion are detrimental to the cell [1,2]. While the amount of free zinc ions in the cytosol of most cells is extremely low, it is high in vesicles, such as insulin granules, synaptic vesicles, and cortical granules in the oocyte [3,4,5]. In oocytes, a key component of the cortical granules content is ovastacin, an oocyte-specific zinc metalloendopeptidase (Burkart et al., 2012). Particularly in female reproductive biology, Zn is involved in follicular growth, germ cell development, meiosis arrest, oocyte maturation, fertilization, oocyte activation, and polyspermy block (Reviewed by [6]). 

Several reports have described a rapid exocytosis of zinc from the oocyte during activation [5,7,8,9], generating an increase of a 300% around the zona pellucida of this metal [10]. Interestingly, the amount of zinc released at the time of fertilization/activation correlates with embryo development, with larger spark amplitudes associated with higher quality embryos [11]. This loss of zinc during fertilization induces the exit from metaphase II arrest, and the use of Zn chelators can trigger oocyte activation bypassing Ca oscillations [7,8,12,13]. Among Zn chelators used for this propose, N,N,N′,N′-tetrakis(2-pyridylmethyl)ethylenediamine TPEN [7,12,13,14]) has been mostly used. However, more recently, the novel Zn chelators 1,10-phenanthroline (PHEN) and ris(2-pyridylmethyl)amine have now been tested [13]. One probable mechanism to explain the role of Zn during oocyte activation is that the meiosis inhibitors within the oocyte rely upon cytosolic zinc levels. The mechanism underlying the role of Zn during oocyte activation has yet to be fully understood and may not be identical to the Ca-based activation methods [15].

Artificial-induced oocyte activation has a crucial impact on the production of somatic cell nuclear transfer (SCNT) embryos [16] and for assisting oocyte activation when this phenomenon is not properly triggered by the sperm upon fertilization. Altered sperm-induced Ca signaling at fertilization compromises the blastocyst transcriptome [17], preimplantation, and post-implantation development [18], possibly altering epigenetic events and the expression of key cell differentiation transcription factors [19,20]. However, the mechanism behind this and the precise role of Zn remain to be elucidated. We explored new conditions and treatments for achieving optimal embryo developmental rates, comparing embryo quality and the cell differentiation markers SOX2, CDX2, OCT4, and SOX17 with embryos generated by conventional in vitro fertilization in bovine or by an ionomycin-induced calcium oscillation in bovine and porcine. 

## 2. Materials and Methods

Unless otherwise stated, all chemicals were obtained from Merck KGaA, Darmstadt, Germany.

### 2.1. Ethics and Animal Welfare Statement 

The experiments performed in this manuscript did not require the approval from the Ethics and Animal Welfare Committee of the Faculty of Agriculture, University of Buenos Aires given that no living animals were involved (CICUAL-FAUBA, Res. CD 1476/19, Reglamento para el cuidado y uso de animales para enseñanza, investigación y servicios). 

### 2.2. Bovine Cumulus-Oocyte Complexes Collection and In Vitro Maturation

Bovine ovaries were collected from a local abattoir and transported to the laboratory (2 h transportation time). Cumulus-oocyte complexes (COCs) were aspirated from all small antral follicles ranging from 2 to 8 mm in diameter. Only COCs covered with at least three layers of cumulus cells were selected and maturated in vitro for 22–24 h. The maturation medium was Tissue Culture Medium 199 (31100-035, Gibco, Grand Island, NY, USA) supplemented with 10% *v*/*v* fetal bovine serum (10091148, Thermo Fisher Scientific, Waltham, MA, USA), 100 μM cysteamine (M9768), 0.3 μM sodium pyruvate (P2256), 1% *v*/*v* antibiotic-antimycotic (15240-096; Gibco, Grand Island, NY, USA) and 10 μg/mL follicle-stimulating hormone (NIH-FSH-P1; Folltropin ^®^, Bioniche, Belleville, ON, Canada). COCs were incubated in 100 μL droplets (20–25 COCs/droplet) covered with mineral oil (0121-1; Fisher Chemical, Pittsburgh, PA, USA) at 5% CO_2_ in humidified air at 38.5 °C. After in vitro maturation, COCs were vortexed for 2 min in 1 mg/mL hyaluronidase (H-4272) in Dulbecco’s phosphate saline (DPBS) for cumulus cell removal. The first polar body extrusion was evaluated by direct observation on a stereoscopic microscope (SMZ 800; Nikon, Tokyo, Japan). Matured oocytes were randomly allocated to activation or IVF groups. 

### 2.3. Porcine Cumulus-Oocyte Complexes Collection and In Vitro Maturation

Ovaries were collected from a local slaughterhouse and transported to the laboratory at around 25 to 30 °C within 3 h of collection. COCs from follicles 3–6 mm in diameter were aspirated using an 18-gauge needle attached to a 10 mL disposable syringe. Compact COCs were selected and matured in 100 µL droplets of tissue culture medium bicarbonate-buffered TCM-199 under mineral oil, supplemented with 0.3 mM sodium pyruvate, 100 mM cysteamine, 5 μg/mL myo-inositol, 1 µg/mL insulin-transferrin-selenium, 1% *v*/*v* antibiotic-antimycotic, 10% *v*/*v* porcine follicular fluid (follicular fluid was obtained from follicles of 3–6 mm of diameter, centrifuged at 1900× *g* for 30 min at 5 °C, filtered and then aliquoted and stored at −20 °C), 5 ng/mL basic Fibroblast Growth Factor (F3685), and 10 µg/mL of FSH. Maturation was performed at 38.5 °C in a humidified atmosphere of 5% CO_2_ in 90% air for 42–44 h.

### 2.4. Bovine Oocyte Artificial Activation

Selected bovine oocytes were randomly allocated to different treatments in each experiment according to the experimental groups. Ionomycin was used as a control group for experiments 1 and 2:5 μM ionomycin (I24222, Thermo Fisher Scientific, Waltham, MA USA) in Talp-h for 4 min at 38.5 °C in the dark. For experiment 1, 0.5 mM PHEN (P9375) in Talp-h for 1 h or 0.5 mM PHEN in synthetic oviductal fluid (SOF, [21] without fetal bovine serum) for 1 h at 38.5 °C in the dark were used. For experiment 2, PHEN treatments were performed in Talp-h media for 30 min or 1 h at 0.5 mM or 1 mM at 38.5 °C in the dark. After incubation in Ionomycin or PHEN, oocytes were washed in Talp-h for 5 min and imminently cultured for 4 h at 38.5 in a 100 µL drop of IVC media containing 1.9 mM 6-dimethylaminopurine (D2629) covered with mineral oil. After incubation, presumptive zygotes were washed in Talp-h for 10 min and placed in calibrated SOF media for in vitro culture. 

### 2.5. Porcine Oocyte Artificial Activation

Selected porcine oocytes were randomly allocated to different treatments in each experiment according to the experimental groups. Ionomycin used as control group in all experiments: 5 μM ionomycin (I24222, Thermo Fisher Scientific, Waltham, MA USA) in Talp-h for 4 min at 38.5 °C in the dark. For experiment 1, 0.5 mM PHEN in Talp-h for 1 h or 0.5 mM PHEN in porcine zygote medium (PZM, [22]) for 1 h at 38.5 °C in the dark were used. Similar to bovine, for experiment 2, PHEN treatments were performed in Talp-h media for 30 min or 1 h incubation at 0.5 mM or 1 mM concentration at 38.5 °C in the dark. For experiment 4, PHEN 1 h for 30 min alone, followed by Ionomycin treatment, and vice-versa were tested as activation protocols.

### 2.6. Bovine In Vitro Fertilization

For experiment 2, bovine COCs were subjected to in vitro fertilization as another control group for this species. Briefly, frozen semen was thawed in a 37 °C water bath for 30 s. Sperm were washed twice by centrifugation at 490× *g* for 5 min with Brackett’s defined medium [23]. Sperm concentration was adjusted to 15 × 10^6^/mL in Brackett’s fertilization medium and co-incubated for 5 h with groups of 20–25 COCs. Afterward, presumptive zygotes were vortexed for 30–60 s, washed several times in Talp-h, and cultured in vitro as described below.

### 2.7. In Vitro Culture of IVF and Parthenogenetic Bovine Embryos

Presumptive IVF and parthenotes bovine zygotes were cultured in SOF supplemented with 2.5% fetal bovine serum for 7 days. Embryos were cultured in a humidified gas mixture of 5% CO_2_, 5% O_2,_ and 90% N_2_ at 38.5 °C. Cleavage and blastocyst rates were recorded on days 2 and 7, respectively. On day 5, 10% of FBS was added to culture media. On day 7, blastocysts were fixed for immunofluorescence analysis. 

### 2.8. In Vitro Culture of Parthenogenetic Porcine Embryos

Presumptive parthenotes zygotes were cultured in PZM media droplets for 7 days. Embryos were cultured in a humidified gas mixture of 5% CO_2_, 5% O_2,_ and 90% N_2_ at 38.5 °C. Cleavage and blastocyst rates were recorded on days 2 and 7, respectively. On day 5, 100% of the media was renewed. On day 7, blastocysts were fixed for immunofluorescence analysis. 

### 2.9. Blastocyst Fixation, Immunofluorescence, and Cell Counting 

Blastocysts were fixed for 20 min in 4% formaldehyde (50980487, Electron Microscopy Sciences, Hatfield, PA USA) in DPBS, rinsed in DPBS with 0.4% *w*/*v* bovine serum albumin (BSA, A6003), and stored at 4 °C in 1 mL tubes. Embryos were treated with permeabilization solution (DPBS containing 0.2% *v*/*v* Triton X-100 (21123)) for 15 min and washed in blocking buffer (DPBS containing 0.1% *v*/*v* Tween 20 (P9416) and 0.4% *w*/*v* BSA). A negative control group without primary antibody was included for all assays. Fixed blastocysts were incubated with SOX2 antibody (Sox2 (D9B8N), 1:200, rabbit monoclonal, Cell Signaling, #23064) overnight at 4°C in blocking buffer. Afterward, bovine embryos were washed three times for 15 min in blocking buffer, followed by 1 h incubation at room temperature with CDX2 and SOX17 antibody (CDX2-88 antibody, 1:100, mouse monoclonal, Biogenex, AM392GP; SOX17 antibody, 1:100, Polyclonal Goat IgG, R&Dsystem, AF1924). Porcine blastocysts were incubated for 1 h at room temperature with CDX2 and OCT4 antibodies (Oct-3/4 Antibody (N-19), 1:100, goat polyclonal, sc-8628). After washing, embryos were incubated with secondary antibodies (Alexa Fluor^®^ 555, 1:1000, donkey anti-goat, #21432, Alexa Fluor^®^ 488, 1:1000, donkey anti-mouse IgG #A21202, Alexa Fluor^®^ 488, 1:1000, donkey anti-rabbit, #A-31573, Thermo Fisher Scientific, Waltham, MA USA) in blocking buffer for 1 h at room temperature in the dark. Finally, blastocysts were mounted in Vectashield^®^ containing 1.5 µg/mL DAPI (Vector Laboratories, Burlingame, CA, USA), and slides were scanned using an inverted confocal microscope (Olympus IX83 Spinning Disk Confocal System). The analysis of total cell number and quantification of SOX2 positive (SOX2+), SOX17 positive (SOX17+), OCT4 positive (OCT4+), and CDX2 (CDX2+) nuclei was performed manually with FIJI image processing software [24]. For image analysis, we use depth of nuclei information within an embryo to facilitate accurate cell counting as described by [25].

### 2.10. Statistical Analyses

All statistical analyses were performed using GraphPad Prism software version 9.4.1. Comparisons of three groups were performed using Kruskal–Wallis test with Dunn’s multiple comparisons test. For analysis of two groups, Mann–Whitney test was used. Differences were considered statistically significant with a value of *p* ≤ 0.05. A minimum of three biological replicates were performed and analyzed for each experiment.

## 3. Results

### 3.1. Experiment 1: Effect of Two Culture Media on Bovine and Porcine Oocyte Activation Rates and Embryo Development after Oocyte Activation with PHEN

To first assess whether the media in which oocytes are exposed to Zn chelator influences the success of activation and embryo development, we compared a previously published protocol for porcine oocytes [13] using PHEN in two different media: Talp-h or SOF for bovine and Talp-h or PZM for porcine. In bovine, PHEN parthenogenetic activation in Talp-h resulted in better cleavage rates but did not significantly influence blastocyst developmental rates relative to the use of SOF. Moreover, regardless of the culture media used, egg activation using the Zn chelator resulted in significantly reduced developmental rates compared to ionomycin treatment. In pigs, PHEN activation in Talp-h supported better developmental rates than PZM. Results are shown in Table 1 and Table 2. Talp-h was used as the media for PHEN activation in experiments 2 and 4 for both species. 

### 3.2. Experiment 2: Effects of Different PHEN Concentrations and Incubation Times on Bovine and Porcine Activation Rates and Parthenogenetic Developmental Competence

After establishing that Talp-h was a more suitable media for activation using the Zn chelator PHEN, we explored different concentrations and incubation times to improve parthenogenetic activation rates in bovine and porcine oocytes. As controls, bovine IVF and parthenogenetic embryos generated by ionomycin were included. In bovine, compared to Ionomycin and IVF controls, development to blastocyst was compromised in all PHEN experimental groups. Although not significantly different from other PHEN groups, the highest cleavage and blastocyst rate among PHEN groups was 0.5 mM for 1 h. Results are shown in Table 3. In porcine, only Ionomycin control was included. The experimental group of PHEN 1 mM for 30 min showed significantly higher developmental rates than all other experimental groups, including the Ionomycin control (Table 4).

### 3.3. Experiment 3: Effects of Zn Chelation on the Quality and Cell Differentiation Markers Expression of Bovine and Porcine Blastocysts 

To understand the impact on embryo quality and early cell differentiation of in vitro embryos generated by different treatments, we performed immunofluorescence to assess total cell number and the expression of CDX2, SOX2, and SOX17 transcription factors in bovine and CDX2, SOX2, and OCT4 in porcine. Bovine blastocyst produced with PHEN 0.5 mM for 1 h showed a significantly lower total cell number compared to IVF (mean ± SEM, IVF 116.5 ± 7.56; Ionomycin, 91.00 ± 7.30; PHEN 85.19 ± 5.16). Moreover, bovine PHEN blastocysts showed more SOX2+ cells (52.70 ± 4.5) than IVF embryos (29.80 ± 4.5). Interestingly, a significantly higher proportion of the bovine PHEN embryos (40.00%, eight out of 20) showed a scattered pattern of SOX2 expression (no clear ICM) compared to less than 5.56% (one out of 18) and 5.56% (one out of 18) for Ionomycin and IVF groups, respectively. No differences among bovine experimental groups were found in SOX17 or CDX2 cell numbers (Figure 1 and Figure 2). For porcine, no significant differences were found in total cell number or the expression of any studied marker (Figure 3 and Figure 4).

### 3.4. Experiment 4: Effects of Combining Zn Chelation and Calcium Rise and Vice-Versa on Porcine Preimplantation Embryo Development

To explore the consequences of depleting oocytes of zinc before the artificial induction of a calcium rise or the opposite, we combined the treatments Ionomycin and PHEN (PHEN in Talp-h for 1 mM for 30 min), and we recorded the developmental rates after embryo culture in porcine. Interestingly, regardless of the order, the combination of treatments impaired developmental competence compared to the use of PHEN alone (Table 5). 

## 4. Discussion

Artificial activation methods are suboptimal as they cannot emulate the natural signaling pathway used by the fertilizing sperm. Therefore, understanding the mechanisms behind oocyte activation is crucial for improving in vitro developmental procedures across different mammals, including humans. Failures in normal oocyte activation have been associated with alteration in preimplantation embryo development [26] and offspring health [17]. The results of our study are particularly relevant to determine the role that Zn plays in oocyte activation in different domestic species of importance for production and biomedicine.

Bovine oocytes release Zn in response to ionomycin-induced and sperm-induced egg activation [5]. In pigs and mice, Zn chelation triggers activation and supports preimplantation development in vitro [7,27]. These lead us to hypothesize that there could be similar mechanisms of Zn-mediated activation for the bovine. Here, we provide the first insights into parthenogenetic blastocyst development in cattle using Zn chelation. In this species, assisted oocyte activation has been used to generate haploid and diploid parthenogenetic [28] and androgenetic embryos [29]. Moreover, it is an essential step in SCNT and for assisting round spermatid sperm injection. Interestingly, despite the recent advances in the area, the efficiency of SCNT in terms of healthy newborns in mammalian species is still very low, and a better oocyte activation protocol could improve cloning efficiency [16]. While IVF has been performed successfully in bovine, [30], ICSI bovine and porcine embryos fail to develop in vitro, and low blastocyst rates are obtained [31]. However, this can be improved by assisted activation [32,33,34]. Our work demonstrated that bovine oocytes activated by using a Zn chelator and, bypassing Ca oscillations [13], can develop up to the blastocyst stage, but the developmental rates and quality of these embryos are compromised. On the other hand, porcine oocytes showed to be easily activated by zinc chelation compared to bovine oocytes. Thus, we demonstrated that the efficiency of oocyte activation by inducing an artificial Zn depletion depends on the species. This could be related to an inequality response to the chelator, different levels of Zn accumulated during oocyte maturation, or species-specific activation signaling pathways. Further research is needed to elucidate these assumptions. 

As mentioned, bovine blastocysts obtained using only Zn chelator in our study were compromised in quantity and quality. Thus, in this species, calcium rise during oocyte activation seems important for blastocyst development and early cell differentiation during the preimplantation period. This agrees with a recent study that also suggests a crucial role of calcium in the reprogramming of bovine oocytes [35]. On the contrary, porcine embryos developmental rates were higher and no differences in quality and pluripotent markers were observed. This observation could suggest that Ca oscillations in pigs might induce oocyte activation largely through the mechanism triggered by the Zn sparks. Although the use of Zn chelator has not yet been approved in humans, other assisted artificial activation protocols have been approved [36,37]. Thus, outlining better activation procedures across different mammalian models is essential before application to humans. For this reason, we also investigated the outcomes of combining ionomycin and PHEN treatments in pig oocytes. In our study, using the two treatments in any order did not show any beneficial effect compared to PHEN treatment alone. Moreover, as demonstrated by Ca oscillation alteration using knout mice models [38], disbalances of Zn levels during oocyte activation could impact not only preimplantation embryo development, but also the health of the pregnancies and newborns. More studies are needed to deeply understand the potential benefits of combing treatments, also investigating reducing incubation times with an adequate Zn chelator concentration to better simulate the Zn spark event. 

Zinc is the most abundant transition metal in mouse oocytes and intracytoplasmic levels increases during maturation [5,39]. In bovine, it was observed that zinc is being sequestrated during IVM, whereas zinc supplementation impacted the chromatin state by reducing the level of global DNA methylation, which is consistent with the increased transcription [40]. Thus, maternal zinc plays an important role during the oocyte-to-embryo transition [41,42], when essential processes occur for proper embryo development [43]. Bovine PHEN blastocyst had reduced cell number compared with IVF or Ionomycin. SOX2 is specifically expressed in the ICM and CDX2 represses the expression of SOX2 in the trophectoderm in the cattle [44]. Yet, we detected a significantly higher proportion of nuclei co-expressing CDX2 and SOX2 in PHEN blastocyst, suggesting that this repression mechanism could be altered in this group. On the contrary, no differences in porcine embryos were found in any transcription factors between groups. This agrees with a previous report in mice where the gene expression profile for pluripotency markers in blastocysts produced by Zn chelation was similar to blastocysts generated through a Ca-based activation [7]. Altogether, our results highlight species-specific features of the role of Zn in oocyte activation and the subsequent events during preimplantation embryo development and cell differentiation.

## 5. Conclusions

In summary, our observations suggest that even though in vitro blastocyst development can be achieved by using a Zn chelator in bovine bypassing Ca oscillations, developmental rates and blastocyst quality, particularly SOX2+ cell number and distribution, are compromised compared to embryos generated with sperm-induced calcium oscillations. On the contrary, Zn chelation seems to be sufficient to induce oocyte activation in porcine, and we found an optimal incubation/concentration exposure to PHEN for this species. The combination of treatments for Zn chelation and a Ca rise in porcine oocytes at the concentration tested do not improve developmental competence. Our results contribute to understanding the role of Zn and Ca during oocyte activation and preimplantation embryo development across different mammalian species. 

## Figures and Tables

**Figure 1 animals-12-03560-f001:**
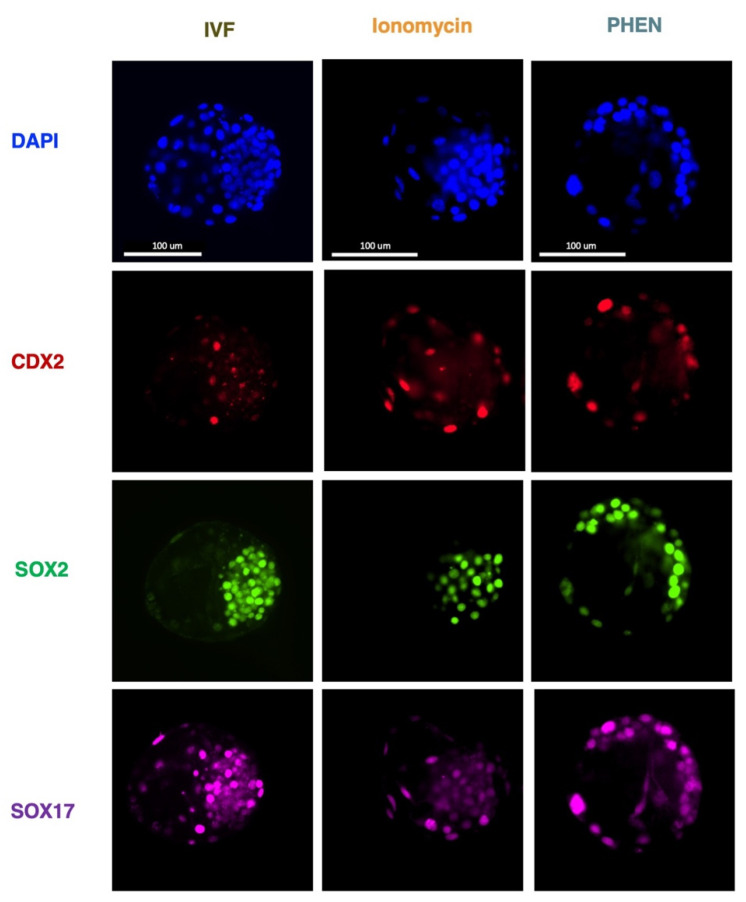
Representative immunofluorescent staining of bovine blastocyst stage embryos of the indicated groups. Scale bars indicate 100 μm. IVF, in vitro fertilization; PHEN, 1,10-phenanthroline.

**Figure 2 animals-12-03560-f002:**
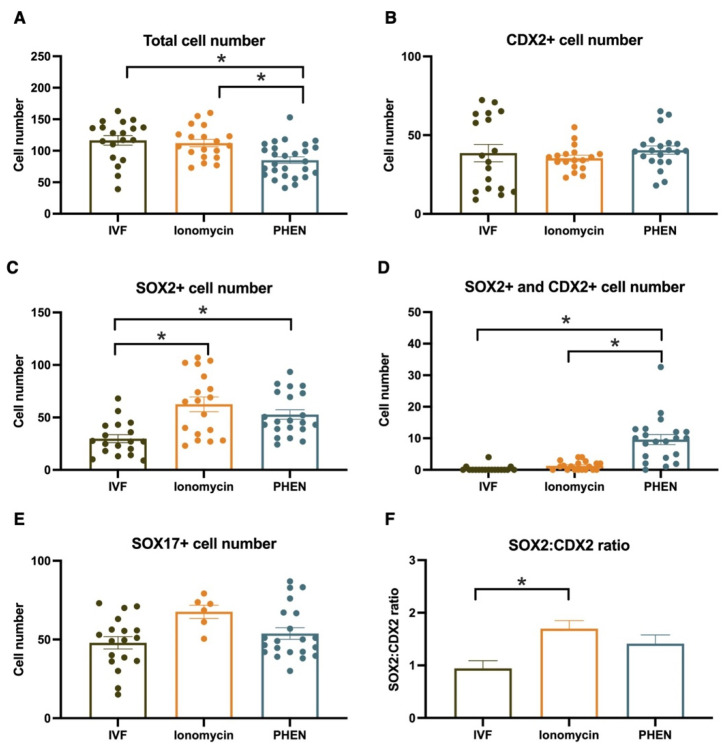
Analysis of cell number and expression of transcription factors of bovine blastocyst from different experimental groups. Blastocysts from three biological replicates were analyzed; 15 to 27 embryos/group; * *p* < 0.05, Kruskal-Wallis test. IVF, in vitro fertilization; PHEN, 1,10-phenanthroline.

**Figure 3 animals-12-03560-f003:**
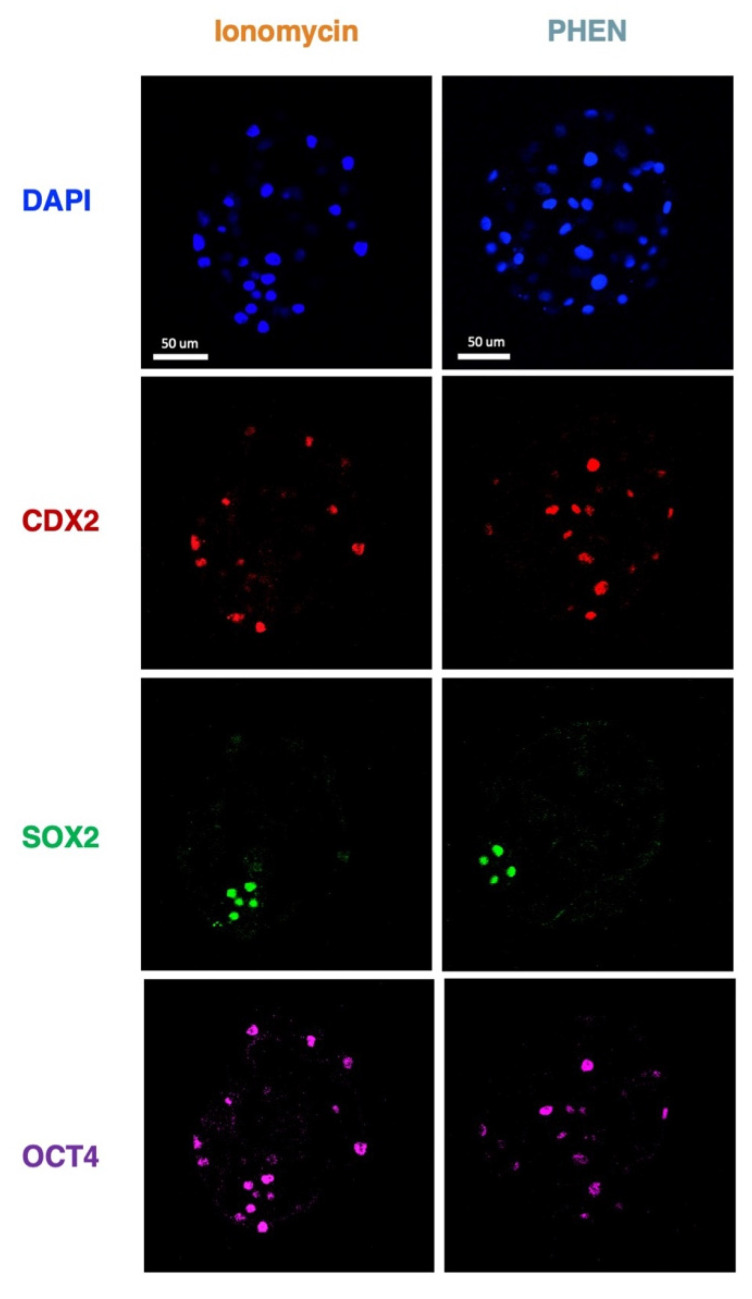
Representative immunofluorescent staining of porcine blastocyst stage embryos of the indicated groups. Scale bars indicate 50 μm. PHEN, 1,10-phenanthroline.

**Figure 4 animals-12-03560-f004:**
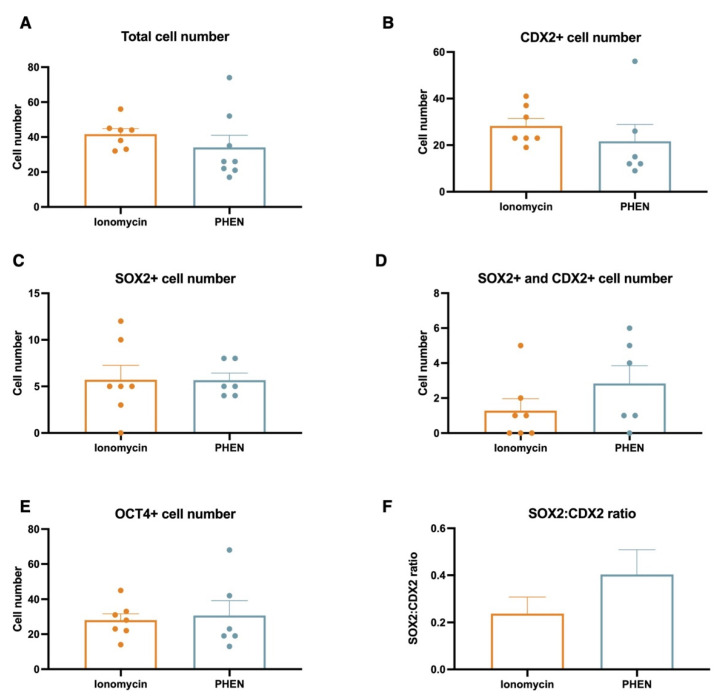
Analysis of cell number and expression of transcription factors of porcine blastocyst from different experimental groups. Blastocysts from three biological replicates were analyzed; 6 to 8 embryos/group; *p* < 0.05, Mann-Whitney test. No statistical differences were found among groups. PHEN, 1,10-phenanthroline.

**Table 1 animals-12-03560-t001:** In vitro preimplantation development of parthenogenetic bovine embryos using the Zn chelator PHEN at 0.5 mM for 1 h in two different media.

Group	n. oocytes	n. cleaved embryos (%)	n. d 8 blastocysts (%)
Ionomycin	80	76 (95.00) ^a^	61 (76.25) ^a^
PHEN–SOF	73	45 (61.64) ^b^	15 (20.55) ^b^
PHEN–Talp-h	93	71 (76.34) ^c^	27 (29.03) ^b^

Different superscripts indicate statistical significance (Kruskal-Wallis with Dunn’s, *p*-value < 0.05). Three biological replicates were performed. PHEN, 1,10-phenanthroline; SOF, synthetic oviductal fluid; Talp-h, Tyrode’s albumin lactate pyruvate-hepes.

**Table 2 animals-12-03560-t002:** In vitro preimplantation development of parthenogenetic porcine embryos using Zn chelator PHEN at 0.5 mM for 1 h in two different media.

Group	n. oocytes	n. cleaved embryos (%)	n. d 8 blastocysts (%)
Ionomycin	62	29 (46.77)	11 (17.74) ^a^
PHEN–PZM	92	47 (51.09)	3 (3.26) ^b^
PHEN–Talp-h	93	64 (68.82)	14 (15.05) ^a^

Different superscripts indicate statistical significance (Kruskal-Wallis with Dunn’s, *p*-value < 0.05). Three biological replicates were performed. PHEN, 1,10-phenanthroline; PZM, porcine zygote medium; Talp-h, Tyrode’s albumin lactate pyruvate-hepes.

**Table 3 animals-12-03560-t003:** In vitro preimplantation development of parthenogenetic bovine embryos using the Zn chelator PHEN in Talp-h media at different incubation conditions.

Group	n. oocytes	n. cleaved embryos (%)	n. d 8 blastocysts (%)
IVF	101	69 (68.31) ^a^	42 (41.58) ^a^
Ionomycin	83	69 (83.13) ^b^	42 (50.60) ^a^
PHEN 0.5 mM for 30 min	73	45 (61.64) ^a^	15 (20,55) ^b^
PHEN 0.5 mM for 1 h	82	62 (75.60) ^a,b^	23 (28.04) ^b^
PHEN 1 mM for 30 min	72	39 (54.17) ^a,c^	20 (27.78) ^b^
PHEN 1 mM for 1 h	72	28 (38.89) ^c^	14 (19.44) ^b^

Different superscripts indicate statistical significance (Kruskal-Wallis with Dunn’s, *p*-value < 0.05). Three biological replicates were performed. IVF, in vitro fertilization; PHEN, 1,10-phenanthroline.

**Table 4 animals-12-03560-t004:** In vitro preimplantation development of parthenogenetic porcine embryos using Zn chelator PHEN in Talp-h media at different incubation conditions.

Group	n. oocytes	n. cleaved embryos (%)	n. d 8 blastocysts (%)
Ionomycin	97	71 (73.20) ^a^	20 (20.62) ^a^
PHEN 0.5 mM for 1 h	146	111 (76.03) ^a^	27 (18.49) ^a^
PHEN 1 mM for 30 min	85	70 (82.35) ^a^	37 (44.71) ^b^
PHEN 1 mM for 1 h	63	31 (49.21) ^b^	9 (14.29) ^a^

Different superscripts indicate statistical significance (Kruskal-Wallis with Dunn’s, *p*-value < 0.05). Three biological replicates were performed. PHEN, 1,10-phenanthroline.

**Table 5 animals-12-03560-t005:** In vitro preimplantation development of parthenogenetic porcine embryos using Zn chelator PHEN or Ionomycin.

Group	n. oocytes	n. cleaved embryos (%)	n. d 8 blastocysts (%)
Ionomycin	79	65 (82.28)	22 (27.85) ^a^
PHEN	76	69 (90.79)	35 (46.05) ^b^
Ionomycin + PHEN	103	85 (82.52)	32 (31.07) ^a^
PHEN + Ionomycin	76	69 (90.79)	16 (21.05) ^a^

Different superscripts indicate statistical significance (Kruskal-Wallis with Dunn’s, *p*-value < 0.05). Three biological replicates were performed. PHEN, 1,10-phenanthroline.

## Data Availability

The data presented in this study are available on request from the corresponding author.

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
