# Peer review of "Preimplantation Developmental Competence of Bovine and Porcine Oocytes Activated by Zinc Chelation"

_animals, 2022, doi:10.3390/ani12243560_

Round 1
Reviewer 1 Report
Cabeza and colleagues tested the Zinc chelator PHEN as an alternative to calcium-mobilizing ionomycin for oocyte activation in cattle and swine. The work is sound and provides novel findings on the topic.
The main limitation is the small number of oocytes per group/experiment and the lack of details on the number of experimental replicates. This fact leads to unusual and discordant results (Table 1 has a 76% blastocyst rate for the control, with inconsistencies between similar experiments in Tables 2 and 5, respectively). These details are critical for making a better assessment of the work.
Major points
- Authors must clearly describe in the introduction what are the differences between their work and previous studies (Uh et al., 2019);
- The conclusion should focus on the main findings, particularly the data on swine.
Minot points
Abstract
mentioning "experiment 1, experiment 2, and so on) does not contribute to the text and should be removed;
Lines 33-36 seem unnecessary and should be deleted;
Introduction
Line 48: replace "last cell" by "In eggs, the..."
Material and Methods
Line 157: How much media was replanished?
Line 181: There is typo here
Results
Tables are not self-explenatory. Insert the missing deatils on group lables, replace "#" by "n", inset the number of experimental replicates, desription of acronyms, etc
Not all references cited in the text were in the reference section and vice-versa;
Tables 1 and 2 are much similar and should become one; The same comment to Tables 3 and 4 (merging them);
Figure 2 needs the description of exatc P values;
Figure 4 must mention that there was no estatistical differences among experimental groups.
Discussion
Lines 272: ...health of newborns
Lines 288-289: ...all eggs of this work were expected to have released...
Author Response
Reviewer 1:
Cabeza and colleagues tested the Zinc chelator PHEN as an alternative to calcium-mobilizing ionomycin for oocyte activation in cattle and swine. The work is sound and provides novel findings on the topic.
The main limitation is the small number of oocytes per group/experiment and the lack of details on the number of experimental replicates. This fact leads to unusual and discordant results (Table 1 has a 76% blastocyst rate for the control, with inconsistencies between similar experiments in Tables 2 and 5, respectively). These details are critical for making a better assessment of the work.
Response: Thank you for the suggestions. The number of biological replicates is now included in the text, tables, and figure legends. Table 1 correspond to bovine embryo development whereas Table 2 is showing the data obtained for porcine. That is the reason why the numbers are different. We have modified the manuscript to clarify this.
Major points
- Authors must clearly describe in the introduction what are the differences between their work and previous studies (Uh et al., 2019);
Response: The introduction has been modified. Uh et al., 2019 only used PHEN in porcine. We used PHEN for porcine and bovine, and we used new different concentrations and incubation times that improved developmental rates. These findings have been highlighted in the manuscript.
- The conclusion should focus on the main findings, particularly the data on swine.
Response: The conclusion of the manuscript has been modified accordingly. However, our report is the first to produce blastocyst in bovine using a zinc chelator. Moreover, the compromised quality and quantity of these blastocysts compared to embryos generated by IVF or Ionomycin is novel. We believe these are finding worth to be mentioned in the conclusions.
Minot points
Abstract
mentioning "experiment 1, experiment 2, and so on) does not contribute to the text and should be removed;
Response: We agree with this comment and the abstract has been re-written.
Lines 33-36 seem unnecessary and should be deleted;
Response: We agree with this comment and the abstract has been re-written.
Introduction
Line 48: replace "last cell" by "In eggs, the..."
Response: The text was modified as suggested.
Material and Methods
Line 157: How much media was replanished?
Response: The text was modified to include this information.
Line 181: There is typo here
Response: The text was modified to correct this mistake.
Results
Tables are not self-explenatory. Insert the missing deatils on group lables, replace "#" by "n", inset the number of experimental replicates, desription of acronyms, etc
Response: These details are now included in tables and figures.
Not all references cited in the text were in the reference section and vice-versa;
Tables 1 and 2 are much similar and should become one; The same comment to Tables 3 and 4 (merging them);
Response: References have now been checked and modified. Tables 1 and 2 belong to different species. We believe it is less confusing for the reader to have the information separated by species. Same as Tables 3 and 4.
Figure 2 needs the description of exatc P values;
Response: The figure legend has been modified following the suggestion of the reviewer.
Figure 4 must mention that there was no estatistical differences among experimental groups.
Response: The figure legend has been modified following the suggestion of the reviewer.
Discussion
Lines 272: ...health of newborns
Response: The text was modified following the suggestion of the reviewer.
Lines 288-289: ...all eggs of this work were expected to have released...
Response: The text was modified following the suggestion of the reviewer.
Reviewer 2 Report
I evaluate the manuscript, and I showed my comments and request in attached file. First of all, in title 'oocyte' should be used instead of 'egg' since this word sounds more scientific. I recommend to use 'oocyte' as much as possible in text as well.
In keywords, genetical terms should be excluded because it is out of main objective and there is no direct genetical experiment. In key words, 'Zn chelator' should be emphasized.
In material and methods, there is a confusion. According to authors expression, there is one control group. However, Tables in results section, there is one control group in each experiment. Therefore, this mistake should be correctly defined in definition of experimental groups and statistical methods. My understanding is groups (number of groups differs in different experiments) each experiment is separately analyzed. All these confusion should be clarified.
In discussion, different results following Zn chelator in bovine and porcine should be discussed in different paragraph. What are potential factors? Genetic or different expression?
In all manusript, it is better uniform so please use bovine and porcine not bovine and pig.

Author Response
I evaluate the manuscript, and I showed my comments and request in attached file. First of all, in title 'oocyte' should be used instead of 'egg' since this word sounds more scientific. I recommend to use 'oocyte' as much as possible in text as well.
Response: The text was modified following the suggestion of the reviewer. “Egg” word was only used in the new simple summary as could be simply and concisely to the public. The attached file was considered for improving the manuscript. Thank you for the corrections.
In keywords, genetical terms should be excluded because it is out of main objective and there is no direct genetical experiment. In key words, 'Zn chelator' should be emphasized.
Response: Thank you for this suggestion. Keywords have been modified accordingly.
In material and methods, there is a confusion. According to authors expression, there is one control group. However, Tables in results section, there is one control group in each experiment. Therefore, this mistake should be correctly defined in definition of experimental groups and statistical methods. My understanding is groups (number of groups differs in different experiments) each experiment is separately analyzed. All these confusion should be clarified.
Response: Thank you for this suggestion. We have modified the M&M section (2,4; 2.5; 2.6 sections particularly) to make the experiments and experimental groups more clear.
In discussion, different results following Zn chelator in bovine and porcine should be discussed in different paragraph. What are potential factors? Genetic or different expression?
Response: The discussion of the manuscript has been improved and the suggestion of the reviewer was considered and the potential reasons for the differences between the species are now included.
In all manusript, it is better uniform so please use bovine and porcine not bovine and pig.
Response: We agree with this comment and pig was replaced by porcine in the manuscript.
Reviewer 3 Report
Previous research by different authors has shown that maternal zinc plays an important role during oocyte-to-embryo transition. After the fusion of the spermatozoon with the oocyte, rise of calcium ions (Ca2+) within the cytoplasm of the oocyte triggers the release of Zn-ions out of the egg. These coordinated processes probably play an essential role in the activation of the oocyte. To confirm this hypothesis the authors of the present study aimed to determine the optimal condition for induction of the activation of bovine and porcine oocytes using the Zn chelator 1,10-phenanthroline (PHEN) to compare the parthenogenetic developmental rates and embryo quality with bovine IVF, and Ionomycin-induced embryos of both species. In this in vitro study, the authors wanted to gain a better understanding of the role of Zn2+ and Ca2+ iones during oocyte activation and early embryo development in pigs and cows.
The sophisticated techniques used in the different experiments ( including porcine cumulus-oocyte complexes collection and in vitro maturation, bovine egg artificial activation, porcine egg artificial activation, bovine in vitro Fertilization, in vitro culture of IVF and parthenogenetic bovine embryos, in vitro culture of parthenogenetic porcine embryos, blastocyst fixation, immunofluorescence and cell counting) are sound and well described
The several experiments showed some interesting and important results, which will help to understand better the role of Ca- and Zn ions in oocyte activation and demonstrated significant differences between porcine and bovine eggs. Even though blastocyst development can be achieved in vitro using a Zn chelator in bovine bypassing Ca2+ oscillations, developmental rates and blastocyst quality are compromised compared to embryos generated with sperm-induced calcium oscillations. In the porcine Zn chelation seems to be sufficient to induce egg activation and the authors could also work out the optimal incubation/concentration exposure to PHEN.
In summary, the observations obtained in this study suggest that in the bovine blastocyst development in the bovine can be achieved using a Zn2+ chelator bypassing the normally occurring Ca2+ oscillations, development rates and blastocyst quality are distinctly lower compared to embryos after sperm-induced or artificial induced calcium oscillations. In pig oocytes, zinc chelation on the other hand had a no strong influence on the developmental rates and embryo quality.
Author Response
Previous research by different authors has shown that maternal zinc plays an important role during oocyte-to-embryo transition. After the fusion of the spermatozoon with the oocyte, rise of calcium ions (Ca2+) within the cytoplasm of the oocyte triggers the release of Zn-ions out of the egg. These coordinated processes probably play an essential role in the activation of the oocyte. To confirm this hypothesis the authors of the present study aimed to determine the optimal condition for induction of the activation of bovine and porcine oocytes using the Zn chelator 1,10-phenanthroline (PHEN) to compare the parthenogenetic developmental rates and embryo quality with bovine IVF, and Ionomycin-induced embryos of both species. In this in vitro study, the authors wanted to gain a better understanding of the role of Zn2+ and Ca2+ iones during oocyte activation and early embryo development in pigs and cows.
The sophisticated techniques used in the different experiments ( including porcine cumulus-oocyte complexes collection and in vitro maturation, bovine egg artificial activation, porcine egg artificial activation, bovine in vitro Fertilization, in vitro culture of IVF and parthenogenetic bovine embryos, in vitro culture of parthenogenetic porcine embryos, blastocyst fixation, immunofluorescence and cell counting) are sound and well described
The several experiments showed some interesting and important results, which will help to understand better the role of Ca- and Zn ions in oocyte activation and demonstrated significant differences between porcine and bovine eggs. Even though blastocyst development can be achieved in vitro using a Zn chelator in bovine bypassing Ca2+ oscillations, developmental rates and blastocyst quality are compromised compared to embryos generated with sperm-induced calcium oscillations. In the porcine Zn chelation seems to be sufficient to induce egg activation and the authors could also work out the optimal incubation/concentration exposure to PHEN.
In summary, the observations obtained in this study suggest that in the bovine blastocyst development in the bovine can be achieved using a Zn2+ chelator bypassing the normally occurring Ca2+ oscillations, development rates and blastocyst quality are distinctly lower compared to embryos after sperm-induced or artificial induced calcium oscillations. In pig oocytes, zinc chelation on the other hand had a no strong influence on the developmental rates and embryo quality
Response: We truly appreciate the comments of the reviewer. These were considered for improving the discussion section of the manuscript.